# Morphological and Compositional Features of Chronic Internal Carotid Artery Occlusion in MR Vessel Wall Imaging Predict Successful Endovascular Recanalization

**DOI:** 10.3390/diagnostics13010147

**Published:** 2023-01-01

**Authors:** Jin Zhang, Shenghao Ding, Bing Zhao, Beibei Sun, Qinhua Guo, Yaohua Pan, Xiao Li, Lingling Wang, Jianjian Zhang, Jiaqi Tian, Yan Zhou, Jianrong Xu, Chun Yuan, Jieqing Wan, Xihai Zhao, Huilin Zhao

**Affiliations:** 1Department of Radiology, Ren Ji Hospital, Shanghai Jiao Tong University School of Medicine, 160 Pujian Road, Shanghai 200127, China; 2Department of Neurosurgery, Ren Ji Hospital, Shanghai Jiao Tong University School of Medicine, 160 Pujian Road, Shanghai 200127, China; 3Department of Radiology, University of Washington, Seattle, WA 98195, USA; 4 Center for Biomedical Imaging Research, Department of Biomedical Engineering, Tsinghua University School of Medicine, Haidian District, Beijing 100084, China

**Keywords:** carotid artery diseases, arterial occlusive diseases, magnetic resonance imaging, endovascular procedures

## Abstract

Background: We sought to determine if the morphological and compositional features of chronic internal carotid artery occlusion (CICAO), as assessed by MR vessel wall imaging (MR-VWI), initially predict successful endovascular recanalization. Methods: Consecutive patients with CICAO scheduled for endovascular recanalization were recruited. MR-VWI was performed within 1 week prior to surgery for evaluating the following features: proximal stump morphology, extent of occlusion, occlusion with collapse, arterial tortuosity, the presence of hyperintense signals (HIS) and calcification in the occluded C1 segment. Multivariate logistic regression was used to identify features associated with technical success and construct a prediction model. Results: Eighty-three patients were recruited, of which fifty-seven (68.7%) were recanalized successfully. The morphological and compositional characteristics of CICAO were associated with successful recanalization, including occlusions limited to C1 and extensive HIS, as well as the absence of extensive calcification, absence of high tortuosity, and absence of artery collapse. The MR CICAO score that comprised the five predictors showed a high predictive ability (area under the curve: 0.888, *p* < 0.001). Conclusion: the MR-VWI characteristics of CICAO predicted the technical success of endovascular recanalization and may be leveraged for identifying patients with a high probability of successful recanalization.

## 1. Introduction

Chronic internal carotid artery occlusion (CICAO) is not an uncommon finding in screening clinics and is accountable for a substantial portion of large-artery atherosclerotic strokes [1]. Despite optimal medical therapy, patients with CICAO are at a high risk of cerebral infarction and also may suffer accelerated cognitive decline, due to impaired cerebral perfusion [2,3]. Endovascular revascularization could be beneficial for patients with CICAO [4,5,6,7], but is technically challenging, as the occluded segment can be long, tortuous, and/or heterogenous in composition [8,9,10]. A systematic pre-procedural imaging evaluation used to identify suitable patients is, therefore, an important component of clinical care for achieving high success rates and satisfactory patient outcomes.

Digital subtraction angiography (DSA) is currently the gold standard for the diagnosis and pre-procedural assessment of CICAO [11,12,13]. Some studies suggested that stump morphology, occlusion length, and distal hemodynamic conditions of CICAO were closely associated with the success rate of endovascular recanalization [9,14,15]. A prediction model based on DSA assessment (the DSA CAO score) has been proposed by Chen et al. for predicting the success rate of endovascular recanalization [9]. However, DSA has certain inherent limitations, including invasiveness, ionizing radiation exposure, and limited capability in characterizing the characteristics of invisible segments of occlusion (e.g., negative remodeling, fibrosis and intraluminal thrombi).

Recently, rapid three-dimensional (3D) MR vessel wall imaging (MR-VWI) techniques have been developed to visualize both luminal stenosis and vessel wall pathologies, which can diagnose CICAO reliably and assess its morphological and compositional characteristics directly [16,17,18]. In this study, we aimed to evaluate if the morphological and compositional features of CICAO, as assessed by MR-VWI, can predict the success of endovascular revascularization. We further constructed an MR-VWI-based prediction score for easy clinical use.

## 2. Materials and Methods

Patients clinically diagnosed with symptomatic CICAO starting at the cervical segment (C1) by ultrasound, CTA, MRA, or DSA, and scheduled for endovascular recanalization were recruited to this study. The exclusion criteria included the following: (1) occlusions starting at the petrous (C2) or a more distal segment; (2) history of surgery with the internal carotid artery; (3) contralateral carotid artery occlusion; (4) contraindications to MRI or DSA examination; (5) choosing conservative management. The study was approved by the ethics committee of Renji Hospital. Written informed consent was obtained from the study participants.

### 2.1. Carotid MR-VWI

All the participants underwent carotid MR-VWI examination within 1 week before scheduled surgery on a 3.0T whole-body scanner (Achieva TX, Philips Healthcare, the Netherlands), with a dedicated eight-channel phased-array carotid artery coil (Chenguang Technologies, Shanghai, China). The protocol included 3D MERGE (motion-sensitized driven equilibrium prepared rapid gradient echo) [17], a fast large-coverage sequence with isotropic spatial resolution, scanned coronally with the following parameters: repeat time/echo time of 9.3/4.4 ms, flip angle of 6°, field of view of 250 × 160 × 64 mm^3^, spatial resolution of 0.8 × 0.8 × 0.8 mm^3^, and acquisition time of 2 min 42 s. Additionally, the protocol included 3D SNAP (simultaneous non-contrast angiography and intraplaque hemorrhage) [19,20], fast field echo (FFE), TR/TE of 89.9/4.8 ms, flip angle of 11/15°, field of view of 250 × 151 × 60 mm^3^, spatial resolution of 0.8 × 0.8 × 0.8 mm^3^ and scan time of 1 min 59 s. 

Images were imported to a workstation (AW4.6; GE Healthcare) for review by two radiologists (J.Z. and H.Z., with 4 and 10 years of experience, respectively) blinded to clinical information. In cases of disagreement, a third reviewer (J.X., with 20 years of experience) was consulted to reach a consensus. Multi-planar reconstruction (MPR) reformats were generated as necessary during the image review. The following lesion characteristics were evaluated: (a) the extent of the occlusion, which was classified using MERGE as being limited to or beyond the cervical (C1) segment; (b) proximal stump morphology, which was classified using 3D MERGE as tapered, blunt, or absent [18]; (c) hyperintense signals (HIS), representing an intraplaque hemorrhage or fresh thrombus [21,22,23], which was determined on 3D SNAP in the occluded C1 segment and classified as minimal/moderate (<50% of the total vessel area at any cross-section) or extensive (≥50%); (d) calcification, which was identified as hypointense signals on both 3D MERGE and 3D SNAP [20,24] in the occluded C1 segment and classified as minimal/moderate or extensive, similar to HIS; (e) high tortuosity, which was defined as having at least 1 bend of >45° on the MPR view of the 3D MERGE images in the occluded C1 segment [25]; (f) artery collapse, representing severe negative remodeling, which was defined if the total vessel area had a diameter of <50% of that of the contralateral ICA at the same level. Figure 1 illustrates the above MR-VWI characteristics of CICAO.

### 2.2. DSA Assessment

Pre-procedural DSA images were interpreted by a neurosurgeon (J.W.) blinded to the clinical information and MR images. Consistent with the DSA CAO score [9], the following imaging characteristics were recorded: (1) proximal stump morphology (classified as tapered, blunt, or absent); (2) distal ICA reconstitution (whether it was via ipsilateral or contralateral injection); (3) level of distal ICA reconstitution (whether it was at/before the clinoid segment or at the communicating or ophthalmic segment). The DSA CAO score also includes the history of neurologic events [9]. 

### 2.3. Endovascular Recanalization

Carotid endovascular recanalization was performed by a neurosurgeon (J.W.) with 15 years of neuro-interventional experience in a digital angiography unit (Innova 4100; GE Healthcare). Before the carotid recanalization, patients were prescribed aspirin (100 mg) and clopidogrel (75 mg) daily and atorvastatin (20 mg) every night for at least three days. At the same time, blood pressure, glucose level, low-density lipoprotein, and other risk factors of all the patients were monitored and adjusted to suitable levels for intervention. 

All the carotid recanalization procedures were performed under general anesthesia with continuous invasive blood pressure and electrocardiographic monitoring. An 8-F femoral sheath and guiding catheter were used to arrive at the target common carotid artery. A pilot 150 guidewire was used to guide the Echelon 14 microcatheter through the occluded segment to the distal ICA. Microcatheter angiography was immediately performed to confirm if the microcatheter was in the true lumen. The microcatheter was then changed to a properly sized balloon for dilating the occluded artery from the distal to the proximal segment, and suitable self-expanding stents for ICA were deployed to scaffold the occlusions. Angiography was performed again to determine if the forward blood flow of the previously occluded ICA was rebuilt. If stent expansion was not adequate, balloons would be used for post-dilation to decrease the residual stenosis. When the guidewire failed to pass through the occluded segment or failed to reach the distal true lumen of the occluded vessel after repeated attempts, carotid intervention would be stopped. Successful recanalization was defined as a modified Thrombolysis in Cerebral Infarction grade 3 and residual stenosis <50% [26,27].

### 2.4. Statistical Analysis

All the statistical analyses were conducted using SPSS software, version 22.0. Continuous variables were expressed as the mean ± standard deviation, and categorical variables were summarized as counts (percentage). The Mann–Whitney U test and chi-square test were used to compare clinical and imaging characteristics between patients with and without successful endovascular recanalization. Univariate and multivariate logistic regression were used to identify independent predictors for successful recanalization and to build a prediction model with a backward selection procedure. A prediction score was constructed using β coefficients of the predictors, of which the performance was assessed using receiver operating characteristic (ROC) analysis and area under the curve (AUC). A *p*-value of less than 0.05 was defined as statistically significant, and all the *p*-values were two-sided.

## 3. Results

Of the 116 patients recruited to this study from January 2015 to December 2020, 33 were excluded due to the following reasons: (1) conservative management requested by patients (n = 15); (2) occlusions starting at the C2 segment or above (n = 9); (3) poor MR image quality (n = 7); and (4) history of surgery with the index artery (n = 2). The patient selection flow chart is shown in the Appendix A. Of the remaining 83 patients with unilateral CICAO, the mean age was 62.6 ± 7.0 years; 88.0% were male; and 68.7% experienced symptoms in the prior 3 months. The overall technical success rate was 68.7% (57/83). The perioperative complication rate was 10.8% (9/83), and the most common complication, with an incidence rate of 7.2% (6/83), was dissection. No deaths occurred in the perioperative period. The incidence of strokes within 30 days after the procedure was 2.4% (2/83). The clinical characteristics of the success and failure groups are summarized in Table 1.

### 3.1. Imaging Characteristics 

Table 1 also summarizes the MR-VWI and DSA imaging characteristics of CICAO. Compared with those in the failure group, patients in the success group were more likely to have occlusions limited to the C1 segment (70.2% vs. 23.1%, *p* < 0.001) and extensive HIS (54.4% vs. 19.2%, *p* = 0.004) and less likely to have extensive calcification (8.8% vs. 34.6%, *p* = 0.009), high tortuosity (7.0% vs. 34.6%, *p* = 0.003), and artery collapse (35.1% vs. 76.9%, *p* = 0.001). No significant difference was found in occlusion length in the C1 segment or proximal stump morphology between the success and failure group. In terms of DSA imaging characteristics, compared with those in the failure group, patients in the success group had a significantly higher prevalence of distal ICA reconstitution at or before the clinoid segment (66.7% vs. 26.9%, *p* = 0.001) and a significantly lower prevalence of distal ICA reconstitution with contralateral injection (24.6% vs. 57.7%, *p* = 0.006). 

### 3.2. Association between MR-VWI Characteristics of CICAO and Successful Recanalization

In the logistic regression models adjusted for clinical characteristics, occlusions limited to the C1 segment (OR: 13.84, 95% CI: 3.44–55.76, *p* < 0.001), extensive HIS (OR: 3.99, 95% CI: 1.20–13.22, *p* = 0.024), extensive calcification (OR: 0.19, 95% CI: 0.05–0.72, *p* = 0.014), high tortuosity (OR: 0.12, 95% CI: 0.03–0.51, *p* = 0.004), and artery collapse (OR: 0.20, 95% CI: 0.06–0.63, *p* = 0.006) were significantly associated with successful recanalization. Furthermore, the associations between the morphological and compositional characteristics by MR-VWI and successful recanalization changed minimally after adjusting for the morphological predictors by DSA (Table 2).

### 3.3. The MR-VWI ICAO Score

Multivariate analysis with backward selection identified the occlusions limited to the C1 segment, extensive HIS, extensive calcification, high tortuosity and artery collapse as independent MR-VWI predictors for successful endovascular recanalization (Table 3). The MR CICAO score was constructed using the β coefficients of the five predictors in the multivariate model. As the absolute values of the β coefficients were close, 1 point was assigned to each binary variable to maximize its simplicity for clinical use (Table 4). Figure 2 presents the relationship between the MR CICAO score and the success rate of endovascular recanalization. ROC analysis showed that the MR CICAO score had a high predictive ability (AUC: 0.888, 95% CI: 0.816–0.960, *p* < 0.001), with an optimal cut-off value of 3.5, sensitivity of 0.614 and specificity of 0.962. The success rates of endovascular recanalization according to the MR CICAO score are shown in the Appendix A. In the following section, two example cases are provided.

#### 3.3.1. Example Case 1

A 62-year-old man was observed with left upper limb weakness 2 months before CICAO recanalization. Preoperative DSA showed right ICA occlusion (Figure 3A). The CICAO limited to the C1 segment was depicted by carotid 3D-MR-VWI. Severe hyperintense signal lesions can be observed (Figure 3B,D, arrow). Although mild eccentric calcification (Figure 3B,C, triangle) was found, the total MR CICAO score for this lesion was 4. The rate of successful guidewire crossing occlusion to the distal C3 segment was estimated to be very high, according to our MR CICAO score model, and as expected, this CICAO was endovascular recanalized successfully in the later intervention.

#### 3.3.2. Example Case 2

A 58-year-old woman was observed with left lower limb weakness and inability to walk for more than 4 weeks before CICAO recanalization. Preoperative DSA showed right ICA occlusion (Figure 4A). The CICAO that had spread to the C3 segment was detected by carotid 3D MR-VWI. Calcification (Figure 4B,D, triangle) and luminal collapse (Figure 4B,C, arrow) were found in the occluded C1 segment. The total MR CICAO score for this lesion was only 1. The guidewire crossing occlusion to the distal C3 segment was estimated to be very difficult. In the later intervention, the guidewire could not reach the true lumen after repeated attempts (Figure 4C,D). This CICAO turned out to fail the endovascular recanalization.

## 4. Discussion

In patients with cervical CICAO, we investigated the morphological and compositional characteristics of CICAO using MR-VWI as potential predictors for successful endovascular recanalization. We found that compared to those in which endovascular recanalization failed, the patients with successful recanalization had a higher prevalence of occlusions limited to the C1 segment and extensive HIS, as well as a lower prevalence of extensive calcification, high tortuosity and artery collapse in the occluded C1 segment. The associations between MR-VWI predictors and successful recanalization were largely independent of the predictors provided by DSA, indicating the information obtained by MR-VWI may be unique in current practice. We further concluded that the MR CICAO score can be used as a clinical tool to identify patients with a high probability of successful endovascular recanalization.

This study proposed a simple scoring system based on MR-VWI features for predicting the success of endovascular recanalization of CICAO. The morphological characteristics of CICAO, including the extent of occlusion, high tortuosity and artery collapse in the occluded C1 segment, were associated with successful endovascular recanalization. It is intuitive that a long and tortuous occlusion would be difficult for guidewire to cross and is prone to complications, such as vessel injury [28]. However, the success rates were similar between the occlusion lengths of <50 mm and ≥50 mm in previous studies by Chen et al. [9] and Li et al. [29]. We did not measure occlusion length, but classified CICAO according to the involved segments and vessel tortuosity. Notably, vessel tortuosity can be fully appreciated using MR-VWI, but not easily with DSA. Morino et al. reported that, for chronic total coronary artery occlusions, a bending of over 45° was an independent predictor for the failure of endovascular recanalization [25]. Our results supported the claim that vessel tortuosity was also a factor in determining the success of endovascular recanalization of CICAO. In addition, about half of our study population showed artery collapse, or substantial negative remodeling, in the occluded C1 segment, which predicted unsuccessful recanalization. When there is evidence of negative remodeling in the occluded segment, guidewire advances can be hindered, resulting in an increasing chance of vascular perforation and dissection. 

It is well-established that hard components such as calcification may hinder guidewire crossing through chronic arterial occlusions. Calcification has been shown to be a predictor for revascularization failure during coronary interventions with chronic occlusions [25]. Beyond calcification, evidence is lacking on the relationships between other components and recanalization success [9,30], which is likely to be because other components cannot be easily detected with DSA or CTA. Our data showed that an HIS lesion on 3D SNAP, which may represent an intraplaque hemorrhage or fresh thrombus [21,22,23], was associated with successful recanalization. Extensive HIS in the occluded segment may indicate lesions with more soft tissues that make guidewire crossing easier, compared to occlusions with calcification or dense fibrosis.

No significant association was found between proximal stump morphology and successful endovascular recanalization. Previous evidence on the association between stump morphology and the success rate of recanalization has been inconsistent. Chen et al. reported that CICAO with a tapered stump was more likely to be recanalized successfully than those with a blunt stump or no stump [9]. A study by Hasan et al., on the other hand, found that revascularization of CICAO with either a tapered or blunt stump had a success rate of 100%, which was much higher than CICAO with no stump [15]. A possible reason as to why the proximal stump morphology did not affect the recanalization outcomes in our study might be that the MR-VWI available in this study enabled the neurosurgeons to visualize the anatomy of carotid bifurcation, regardless of the stump types. Consequently, neurosurgeons may be able to estimate the optimal entry of the guidewire into CICAO, even in the absence of stumps on DSA images.

The MR-VWI CICAO score proposed in this study could be used as a clinical tool to estimate the probability of technical success of endovascular recanalization in individuals. With a backward selection procedure, multivariate analysis identified the five MR-VWI characteristics as predictors for successful recanalization in a parsimonious model, whereas the DSA characteristics did not further improve the predictive value. Of note, the DSA characteristics are often based on indirect information, such as distal ICA reconstitution, while the MR-VWI characteristics include direct morphological and compositional characteristics of the occluded segment in both the lumen and vessel wall. 

This study had several limitations. First, this is a single-center study with a relatively small sample size. Future studies are needed to validate the predictive value of the MR-VWI characteristics and the proposed score in independent cohorts. Second, this study was focused on the technical success of endovascular recanalization, and therefore did not look at other outcome measures, such as the improvement in cerebral perfusion. Ultimately, the clinical decision-making in identifying suitable CICAO patients for endovascular recanalization depends on understanding the relationship between the patient and imaging characteristics, with the full spectrum of clinical outcomes. Third, the MR-VWI assessment of CICAO was limited to the C1 segment, as the intracranial segments suffer limited signal-to-noise with the carotid surface coil used in this study. Combined imaging of the extra- and intra-cranial carotid arteries may allow a more comprehensive assessment of CICAO. 

In conclusion, the morphological and compositional characteristics of CICAO, as directly assessed by MR-VWI, were found to preliminarily predict the technical success of endovascular recanalization. Non-invasive MR-VWI has the potential to assist clinical decision-making in the management of CICAO. 

## Figures and Tables

**Figure 1 diagnostics-13-00147-f001:**
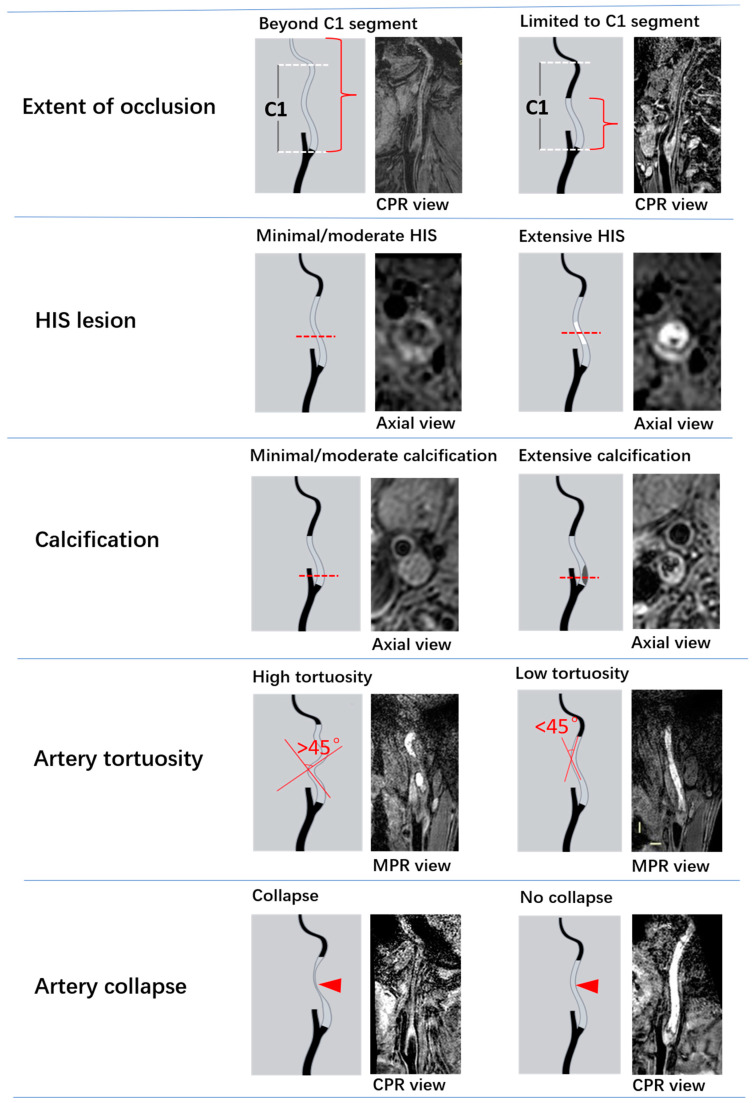
Illustrations of MR-VWI characteristics of CICAO.

**Figure 2 diagnostics-13-00147-f002:**
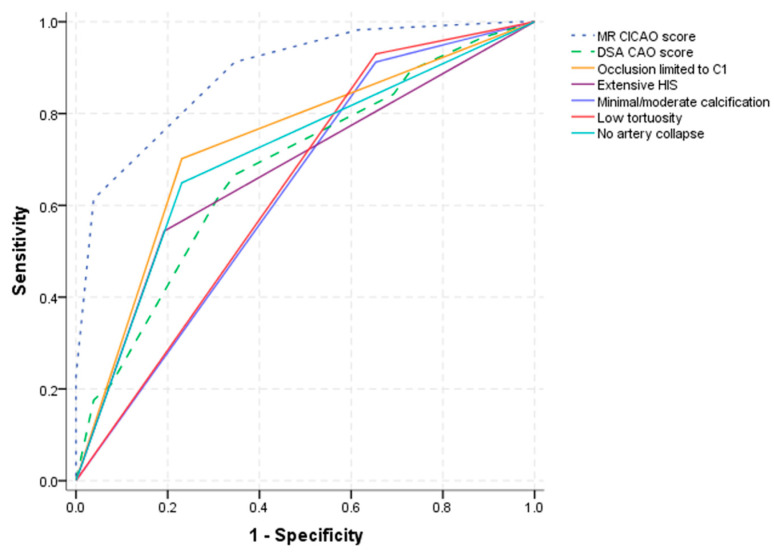
Receiver-operating characteristic (ROC) curves of the MR CICAO score, the DSA CAO score and each single MR-VWI characteristic used to predict successful endovascular recanalization. Area under curve (AUC) of the MR CICAO score = 0.888 (95%CI: 0.816–0.960; *p* < 0.001), cut-off value: 3.5, and AUC of the DSA CAO score = 0.682 (95%CI: 0.559–0.804; *p* = 0.008), cut-off value: 3.5.

**Figure 3 diagnostics-13-00147-f003:**
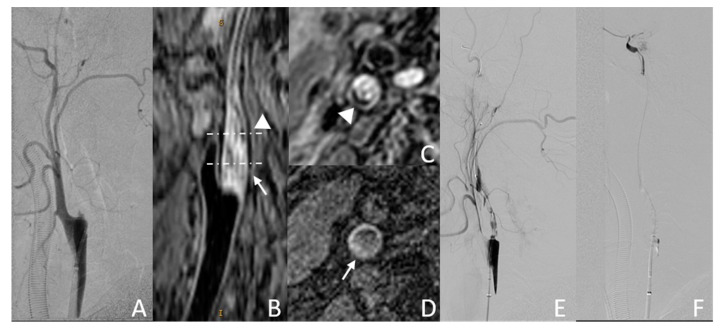
Example case of successful endovascular recanalization. (**A**,**E**,**F**): DSA images; (**B**): MPR image of 3D MERGE; (**C**): axial image of 3D MERGE; (**D**): MPRAGE image. Hyperintense signal lesion ((**B**,**D**), arrow) and some signs of eccentric calcification ((**B**,**C**), triangle) can be observed on MR images.

**Figure 4 diagnostics-13-00147-f004:**
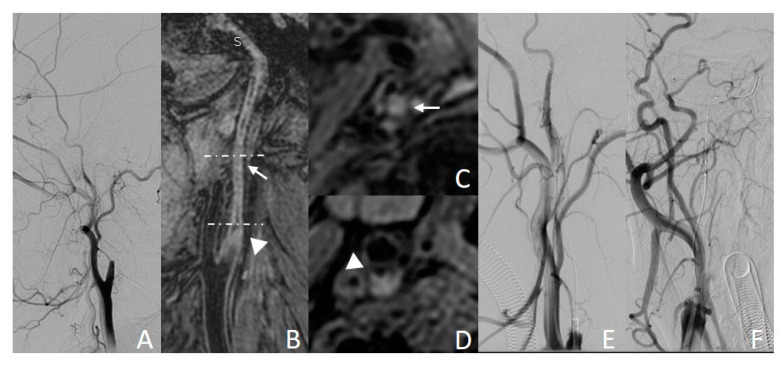
Example case of failed endovascular recanalization. (**A**,**E**,**F**): DSA images; (**B**): MPR image of 3D MERGE; (**C**,**D**): axial images of 3D MERGE. Calcification ((**B**,**D)**, triangle) and collapse ((**B**,**C**), arrow) can be observed on MR images.

**Table 1 diagnostics-13-00147-t001:** Clinical and imaging characteristics.

	All Patients (n = 83)	Success Group (n = 57)	Failure Group (n = 26)	*p* Value
Age (years)	62.6 ± 7.0	62.6 ± 7.1	62.5 ± 6.8	0.945
Male sex	73 (88.0)	53 (93.0)	20 (76.9)	0.064
Smoking	32 (38.6)	26 (45.6)	6 (23.1)	0.057
Alcohol drinking	17 (20.5)	14 (24.6)	3 (11.5)	0.244
Hypertension	59 (71.1)	44 (77.2)	15 (57.7)	0.116
Diabetes	36 (43.4)	28 (49.1)	8 (30.8)	0.154
Dyslipidemia	24 (28.9)	16 (28.1)	8 (30.8)	0.800
Symptom status				0.073
Ischemic event ≤3 months	57 (68.7)	43 (75.4)	14 (53.8)	
Ischemic event >3 months	26 (31.3)	14 (24.6)	12 (46.2)	
Perioperative complication	9 (10.8)	2 (3.5)	7 (26.9)	0.003
Perforation	2 (2.4)	0 (0.0)	2 (7.7)	0.096
Dissection	6 (7.2)	1 (1.8)	5 (19.2)	0.011
Thrombosis	1 (1.2)	1 (1.8)	0 (0.0)	1.000
Stroke within 30 days after procedure	2 (2.4)	1 (1.8)	1 (3.8)	0.531
**MR-VWI characteristics**				
Occlusion length in C1	62.8 ± 23.8	61.7 ± 26.1	65.1 ± 18	0.552
Proximal stump morphology				0.619
Tapered	32 (38.6)	20 (35.1)	12 (46.2)	
Blunt	28 (33.7)	20 (35.1)	8 (30.8)	
Absent	23 (27.7)	17 (29.8)	6 (23.1)	
Occlusions limited to C1	46 (55.4)	40 (70.2)	6 (23.1)	0.001
Extensive HIS	36 (43.4)	31 (54.4)	5 (19.2)	0.004
Extensive calcification	14 (16.9)	5 (8.8)	9 (34.6)	0.009
High tortuosity	13 (15.7)	4 (7.0)	9 (34.6)	0.003
Artery collapse	40 (48.2)	20 (35.1)	20 (76.9)	0.001
**DSA characteristics ***				
Proximal stump morphology				0.519
Tapered	35 (42.2)	22 (38.6)	13 (50.0)	
Blunt	26 (31.3)	18 (31.6)	8 (30.8)	
Absent	22 (26.5)	17 (29.8)	5 (19.2)	
Distal ICA reconstitution with contralateral injection	29 (34.9)	14 (24.6)	15 (57.7)	0.006
Level of distal ICA reconstitution at or before clinoid segment	45 (54.2)	38 (66.7)	7 (26.9)	0.001

Values are % or mean ± SD. Neurologic event was defined as a transient ischemic attack, ischemic stroke or amaurosis fugax. * Based on the DSA CAO score as described by Chen et al.

**Table 2 diagnostics-13-00147-t002:** Association between MR-VWI characteristics of CICAO with successful recanalization.

		Model 1			Model 2	
	OR	95% CI	*p* Value	OR	95% CI	*p* Value
Occlusions limited to C1	13.84	3.44–55.76	<0.001	9.14	2.12–39.34	0.003
Extensive HIS	3.99	1.20–13.22	0.024	4.02	1.09–14.90	0.037
Extensive calcification	0.19	0.05–0.72	0.014	0.31	0.07–1.33	0.115
High tortuosity	0.12	0.03–0.51	0.004	0.18	0.04–0.82	0.027
Artery collapse	0.20	0.06–0.63	0.006	0.23	0.07–0.80	0.021

Model 1: adjusted for clinical characteristics, including age, male sex, smoking and symptom status. Model 2: adjusted for clinical characteristics and DSA characteristics, including distal ICA reconstitution with contralateral injection and distal ICA reconstitution at or before clinoid segment. OR = odds ratio; CI = confidence interval.

**Table 3 diagnostics-13-00147-t003:** Multivariate analysis of MR-VWI predictors of successful recanalization.

	β Coefficient	OR	95% CI	*p* Value
Occlusions limited to C1	1.72	5.56	1.33–23.23	0.019
Extensive HIS	1.55	4.67	1.01–21.84	0.049
Extensive calcification	−1.81	0.16	0.03–0.97	0.046
High tortuosity	−2.37	0.09	0.11–0.79	0.029
Artery collapse	−2.27	0.10	0.20–0.52	0.006

OR = odds ratio; CI = confidence interval.

**Table 4 diagnostics-13-00147-t004:** Variables of the MR CICAO score.

Variable	Categories	Score Point
Extent of occlusion	Beyond the C1 segment	0
	Limited to the C1 segment	1
HIS	Minimal/moderate HIS	0
	Extensive HIS	1
Calcification	Extensive calcification	0
	Minimal/moderate calcification	1
Artery tortuosity	With high tortuosity	0
	With low tortuosity	1
Artery collapse	With collapse	0
	Without collapse	1

## Data Availability

The datasets generated and/or analyzed during the current study are not publicly available due to them containing information that could compromise research participant privacy/consent but are available from the corresponding author [H.Z.] on reasonable request.

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
