# Peer review of "Morphological and Compositional Features of Chronic Internal Carotid Artery Occlusion in MR Vessel Wall Imaging Predict Successful Endovascular Recanalization"

_diagnostics, 2023, doi:10.3390/diagnostics13010147_

Round 1

Reviewer 1 Report

Pre-operative methods for identifying patients suitable for endovascular recanalization are desirable. The endovascular procedure for these patients with CICAO is still challenging, with variable technical success rates. At the same time, DSA cannot depict the wall or lumen of the occluded vessel segment. The authors analyzed the imaging morphological and compositional features of HR-VWI in patients with CICAO and determined their associations with successful recanalization. In the study, eighty-three CICAO patients were included and had an MR-VWI performed within one week before surgery. The logistic regression models found that short occlusion limited to C1, extensive HIS, absence of extensive calcification, absence of high tortuosity, and absence of artery collapse were associated with successful recanalization. The MR CICAO score had a high predictive ability with a sensitivity of 0.614 and a specificity of 0.962. Although the manuscript is well well-written and statistical techniques are sound, the following concerns need to be addressed:

1authors descripted the inclusion and exclusion criteria in the Methods section based on imaging evaluation requirement and only patients scheduled for endovascular recanalization were included, however, the flow chart and the results of patient seems not be consistent to the inclusion and exclusion criteria. Meanwhile, readers might be interesting to know why only 33 of 116 CICAO patients were excluded, as this procedure usually not recommended for most patients.

2) If available, authors are suggested to determine if there are relationships between morphological (high tortuosity and artery collapse) and characteristics of diseased vessel wall.

3) As reflected in the results of the manuscript, the complication rates were higher in the 'failed recanalization' group, were morphological (high tortuosity and artery collapse) and characteristics of diseased vessel wall associated with these complications?

4) Authors didn`t compared the lesion length between the successful and failed treatment group which seems to be important.

6) as DSA was performed in every patient before recanalization, I believed a prediction model combined MR CICAO score with DSA CICAO score would have better diagnostic performance.

7) in the page 2, line 74, “the index carotid artery” should be “the internal carotid artery”

Reviewer 2 Report

This paper seems to be very well organized. The authors should consider the following points

#1 As mentioned in the discussion, the estimated length of occlusion seems to be an important factor in the intervention of completely occluded lesions. Although the classification of lesions is limited to C1 lesions, it is still important to know the estimated length of occlusion. The authors should consider which factor is better, the estimated length of occlusion or the lesion limited to C1.

#2 The authors should explain what you mean by the presence of hyperintense signals, collapses, etc. in the "method" or "discussion" so that readers can understand what you mean.

Reviewer 3 Report

Interesting and well written study regarding Morphological and Compositional Features of Chronic Internal Carotid Artery Occlusion on MR Vessel Wall Imaging Predict Successful Endovascular Recanalization. 

Introduction: clearly presented. 

Material and methods: clear and complete. According to these the study looks like reproducible.

Results: nothing to concern.

Discussion:

- Can you comment about visualization of calcification between MR and CT? Do you think you can detect calcification in MR, or CT remains the gold standard?.

- Did you notice any difference considering the materials which were used during the endovascular treatment ? (eg: Semeraro V et al. Comparison Between Three Commonly Used Large-Bore Aspiration Catheters in Terms of Successful Recanalization and First-Passage Effect. J Stroke Cerebrovasc Dis. 2021 Mar;30(3):105566. doi: 10.1016/j.jstrokecerebrovasdis.2020.105566. Epub 2020 Dec 24. PMID: 33360517. )

Conclusions: should be more cautious. 

Round 2

Reviewer 2 Report

I have no further comments regarding the revised manuscript.